# Efficient Deployment of Transformer Models on Edge TPU Accelerators: A Real System Evaluation

Brendan Reidy, Mohammadreza Mohammadi, Mohammed Elbtity, Ramtin Zand

University of South Carolina. bcreidy@email.sc.edu, mohammm@email.sc.edu, messa@email.sc.edu, ramtin@cse.sc.edu

*Abstract*—**Transformer models have become a dominant architecture in the world of machine learning. From natural language processing to more recent computer vision applications, Transformers have shown remarkable results and established a new state-of-the-art in many domains. However, this increase in performance has come at the cost of ever-increasing model sizes requiring more resources to deploy. Machine learning (ML) models are used in many real-world systems, such as robotics, mobile devices, and Internet of Things (IoT) devices, that require fast inference with low energy consumption. For battery-powered devices, lower energy consumption directly translates into longer battery life. To address these issues, several edge AI accelerators have been developed. Among these, the Coral Edge TPU has shown promising results for image classification while maintaining very low energy consumption. Many of these devices, including the Coral TPU, were originally designed to accelerate convolutional neural networks, making deployment of Transformers challenging. Here, we propose a methodology to deploy Transformers on Edge TPU. We provide extensive latency, power, and energy comparisons among the leading-edge devices and show that our methodology allows for real-time inference of large Transformers while maintaining the lowest power and energy consumption of the leading-edge devices on the market.**

*Index Terms*—**Tensor Processing Unit (TPU), Transformer Models, Edge AI Accelerators, BERT.**

## I. INTRODUCTION

Since the introduction of Transformer models in 2017 [1], they have quickly risen to prominence in many areas, such as natural language processing and computer vision. These models have shown state-of-the-art results in a wide domain of tasks from machine translation [1] and question-answering [2] to computer vision tasks like image segmentation [3]. Many applications, such as self-driving cars, IoT devices, satellites, drones, and robots, require deploying models for real-time inference using low-power energy-constrained systems. Transformer-based models, however, often include a large number of processing layers, along with hundreds of millions of parameters. For instance, the Bidirectional Encoder Representations from Transformers (BERT) [4] models contain 109 million and 340 million parameters for the Base and Large models, respectively [5]. Therefore, deploying such massive models at the edge for real-time applications with tight restrictions on power and energy is challenging.

The surge in demand for specialized hardware for AI applications has resulted in a rapidly expanding industry for edge AI accelerators. Anticipating this trend, several companies have developed their own specialized accelerators. The NVIDIA Jetson Nano [6] is a low-cost development board for machine learning (ML) applications that employ NVIDIA TensorRT1 as the main driver. The Intel Movidius Neural Compute Stick 2 (NCS2) [7] is a small, low-power USB co-processor that enables the deployment of Deep Neural Networks (DNNs) and is powered by the Myriad Vision Processing Unit (VPU). Google's Coral Edge TPU is another device that leverages tensor processing units (TPUs) to accelerate ML applications. The coral TPU is used as a co-processor on Coral's Dev Board, as well as a USB accelerator [8] that can be integrated with tiny computers such as Raspberry Pi. With the peak performance of four tera-operations per second (TOPS) and two TOPS/W, Coral Edge TPU can be one of the promising technologies for realizing real-time Transformer models. While several studies have used the Coral TPU to accelerate their DNN applications, to the best of the authors' knowledge, no work has deployed Transformer-based models on Coral Edge TPU accelerators.

Herein, we propose a methodology to deploy Transformer models on the Coral Edge TPU. Because Transformers are often very large, training them is time-consuming, computationally expensive, and often requires very large datasets that are not always publicly available. For these reasons, it is crucial that our methodology support a wide range of existing Transformer architectures such as Vision Transformers (ViT) [9], left-right Transformers, also known as (a.k.a) Encoder-Decoder Transformers [1], [2] and BERT-like [4] Transformers without any need for retraining, aside from possible retraining associated with quantization. Here, we modify the computational graph to allow the model to run on the Edge TPU while remaining functionally identical to the original model. While common model optimization techniques such as pruning, knowledge distillation, hyper-parameter optimization, and neural architecture search ( [10] provides an overview of these techniques) can be used to improve the size, latency, power consumption, and energy consumption of models, the focus of this paper is on the efficient deployment of existing Transformer architectures on the Coral Edge TPU. Some or all of the aforementioned optimization techniques can be used on top of our work to further improve the latency and power consumption of models. Although we focus on the BERT Transformer architecture for the main body of the work, we show that this methodology can be generalized for both BERT-like and left-right Transformers.

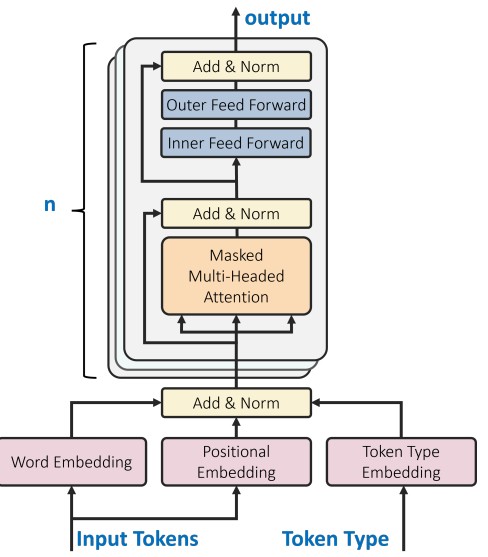

Fig. 1: BERT Architecture with $n$ encoder layers [1].

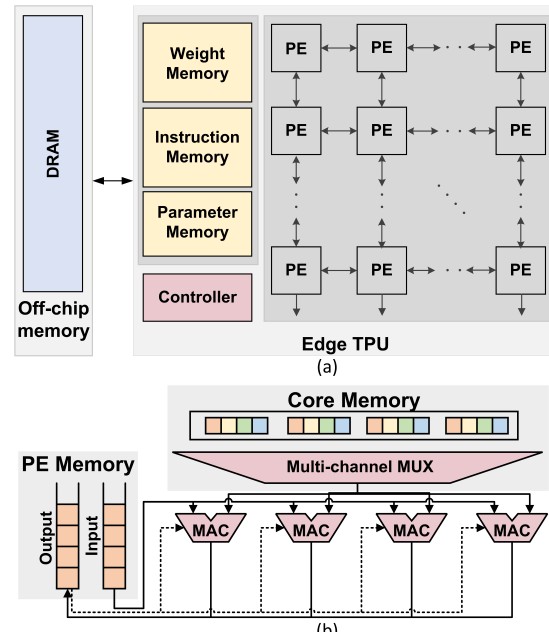

Fig. 2: (a) Edge TPU architecture. (b) PE structure [13].

## II. BACKGROUND

### A. Transformer Model

Transformer models can vary slightly in design, but the core architecture remains the same. Transformers use embedding layers to turn tokens into vectors of size $d_{model}$, a.k.a hidden size. The exact number of embedding layers varies from one model to another. For instance, BERT uses three embedding layers, as shown in Fig. 1. Transformers also employ a stack of attention heads to capture different learned attention associations using scaled dot product attention that maps queries and key-value pairs to outputs. Scaled dot product attention uses a dot product between the queries (Q) and the keys (K) to compute attention scores. These scores are scaled down to create a mean of 0 and a variance of 1, and the Softmax function is applied to generate weights for the values. The weights are then multiplied by the values (V) to generate the weighted attention scores for the tokens. At the end of the multi-headed attention layer, the values from each attention head are concatenated together and passed to a fully-connected (FC) layer and then an activation function is applied.

Most Transformers, including GPT-3 [2], and BERT use Gaussian Error Linear Units (GELU) [11] as the activation function which uses the non-linearity property of Rectified Linear Units (ReLU) with the regularization property of Dropout [12]. The output of the FC layer is added with previous layers using a residual connection. In the Encoder, these values are passed to two FC layers where the inner FC layer has size of $d_{ff}$, a.k.a intermediate size, and the outer FC layer has size of $d_{model}$. Again, the output is added with previous layers and normalized using residual connections. Finally, these values are passed to the next encoder layer, or if there is none, then the classification head/decoder layer. Left-right Transformers have a decoder layer that is nearly identical to the encoder layer, except it has one extra multi-headed attention layer before the feed-forward layers called the encoder-decoder multi-headed attention. The encoder-decoder multi-headed attention is the same as the encoder multi-headed attention except that the query and key vectors come from the encoder, and the values vector comes from the decoder. BERT was introduced in 2018 and builds upon prior Transformer architectures, with one key difference being bi-directionality. Unlike prior Transformer models, BERT is designed to train on both left and right contexts for text. Using a pre-trained BERT model and one additional classification layer, BERT can be fine-tuned to perform various language tasks.

### B. Coral Edge TPU Architecture

In 2015, Google launched the TPU project in which they adopted the systolic array architecture to accelerate the DNN operations [13]. The first version of Google's TPU was designed to only accelerate the DNN inference on the cloud. In 2019, Google launched a smaller and low-power version of TPU, called Edge TPU, that is suited to accelerate the inference of the DNN at the edge. The Edge TPU uses 8-bit integer (int8) multiply and accumulate (MAC) core units in its processing elements (PEs) [8].

In general, the systolic array architecture includes a set of processing elements that are formed in single or multi-dimensional arrays that can collectively perform the computation on certain data brought from memory with no need to access it from the memory multiple times. The systolic arrays developed for the ML acceleration are designed to implement matrix-matrix, matrix-vector, and vector-vector multiplications which are the dominant operations in ML workloads. Systolic arrays increase performance by reusing the values fetched from memory and reducing the main memory accesses [14]. The dataflow in the systolic array is a mapping scheme that depends on the microarchitecture of PEs and determines how

the input data is fed to the array, and how the partial results and outputs are generated and stored. Google adopted the weight stationary dataflow in their cloud TPU and Edge TPU designs [15], in which, the weights are pre-stored in the core memory of PEs. At each cycle, the input elements are fed to the PEs and multiplied by the pinned weights producing partial sums. This process is vertically distributed over columns in the systolic array to produce the output results.

Figure 2 shows the architecture of the Edge TPU and the microarchitecture of each PE within its 2D systolic array. The Edge TPU includes activation memory, instruction memory, parameter memory, controller, and PEs. The controller transfers the data between the off-chip memory and the PEs, fetches parameters and activation into the buffers, and reads the instructions that will be executed on the PEs. The Edge TPU supports a variety of commonly-used operations in DNN models [8]. Each PE in the Edge TPU has four parallel MAC units, as opposed to the cloud TPU v1 which has only one MAC unit per PE. As shown in Fig.2, the PEs in Edge TPU have a single-instruction-multiple-data (SIMD) architecture. They can perform the MAC operation on four data values at the same time using four 8-bit fixed point compute lanes. Moreover, each PE has a core memory and a PE memory. The PE memory is designed as a first in first out (FIFO) buffer that is shared among all PEs and used to store model activations, partial results, and final outputs. Since Edge TPU has a weight-stationary systolic array, the core memory is used to store model parameters, i.e., weights.

## III. PROPOSED METHODOLOGY TO DEPLOY SMALL- AND MEDIUM-SIZED TRANSFORMERS ON EDGE TPU

### A. *Existing Edge TPU Deployment Process*

For full Edge TPU utilization, several requirements must be met; otherwise, only parts of the model will run on the Edge TPU. The Coral documentation [16] contains an exhaustive list of requirements and all supported operations. Here, we only focus on the requirements that are relevant to the Transformer architecture.

The Edge TPU only supports TensorFlow Lite (TFLite) models. TFLite is a lightweight version of TensorFlow [17] that is optimized for deployment on edge systems. Using the TFLite interpreter, different delegates can be used depending on the hardware accelerator, such as NNAPI for android devices, GPU for mobile GPUs, Hexagon for DSPs, Core ML for iOS devices, and *libedgetpu*, which is the focus of this work, for the Coral Edge TPU. Note that TFLite only supports a subset of all TensorFlow operations and the Coral Edge TPU only supports a subset of all TFLite operations. A list of supported Edge TPU operations and any known limitations can be found at [16]. To fully utilize the TPU, the model must contain only supported Edge TPU operations.

Since the Edge TPU only supports 8-bit integer operations, any models aimed to be deployed on Edge TPU must be converted from 32-bit floating point (fp32) to int8 or unsigned int8 for all parameters, activations, and operations. This can be done using either quantization-aware training (QAT) or post-training quantization (PTQ) with a representative dataset. In [18], it is shown that using QAT, BERT can maintain state-of-the-art results using 8-bit integer-only inference. Once the model has been converted to a quantized TFLite model, the Edge TPU compiler maps the supported operations to the TPU and leaves the remaining operations on the CPU. The compiler maps all supported operations onto one graph to be loaded onto the TPU called the *Edge TPU custom op*. Currently, the Edge TPU graph only includes consecutive operations that are supported on Edge TPU. Once the compiler finds an operation in the model that is not supported by the TPU, all the following operations will be mapped to the CPU, regardless of being supported by TPU or not. Another deployment requirement for Edge TPU is that all tensor sizes should be constant at compilation time. After training, we change the batch size dimension to 1 and the sequence length dimension to 128. Moreover, the existing Edge TPU devices do not support embedding layers. Therefore, since the embedding layers make up only a small portion of the overall Transformer model, we leave the operation to run on the CPU for inference.

To verify whether modifying Transformers based on the existing requirements mentioned above would be sufficient to successfully deploy them on Edge TPU, we have adapted BERT-Tiny to BERT-Large models accordingly and tried to deploy them on the Edge TPU. This experiment results in compilation failure or partial compilation for all the models. This is mainly due to the fact that Transformers include operations that are currently not supported by Edge TPU. Thus, we develop several methodologies in the following subsections to resolve the current deployment limitations of Transformers on the Edge TPU.

### B. *Proposed Edge TPU Deployment Process for Transformers*

To address the existing deployment challenges of the Transformers on Edge TPU, it is required to refactor their computational graph to alter their operations to those supported by Edge TPU without altering the model's functionality. Thus we developed a flexible in-house TensorFlow Transformer model using custom Keras layers. This custom Transformer model allows us to modify any operations in our model and replace them where necessary. In order to ensure backward compatibility with existing Transformers, we map pre-trained weights onto our model and verify that both models yield the same output for the same input. In the following, we discuss two of the operations in Transformers that cause the compilation failure in Edge TPU, and propose methods to refactor them such that they can be readily deployed on Edge TPU.

*1) Refactoring GELU Activation Function:* As mentioned, GELU [11] is used in many Transformers and is defined by the following equation:

$$gelu(x) = \frac{1}{2}x[1 + erf(\frac{x}{\sqrt{2}})] \tag{1}$$

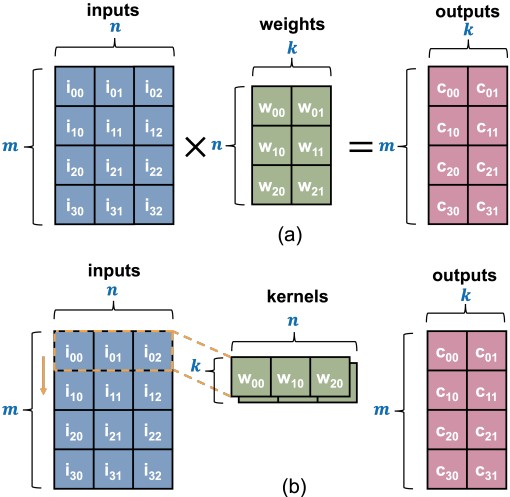

(a)

(b)

Fig. 3: (a) standard matrix-matrix dot product (b) matrix-matrix dot product using convolutions.

TABLE I: Bert models' specifications.

| Model | Hidden size | Attention Heads | Hidden Layers | Intermediate Size | Parameters (millions) |
|---|---|---|---|---|---|
| Tiny | 128 | 2 | 2 | 512 | 4.4 |
| Mini | 256 | 4 | 4 | 1024 | 11.2 |
| Small | 512 | 8 | 4 | 2048 | 28.8 |
| Medium | 512 | 8 | 8 | 2048 | 41.4 |
| Base | 768 | 12 | 12 | 3072 | 109.5 |
| Large | 1024 | 16 | 24 | 4096 | 335.1 |

where $erf(x)$ is the Gaussian error function which is defined as:

$$erf(x) = \frac{2}{\sqrt{\pi}} \int_0^x e^{-t^2} \, dt \tag{2}$$

The GELU activation function is not currently supported on Edge TPU. Several approximations for GELU have been developed, including those based on transcendental functions [11] and those based on polynomial equations [18]. For our purposes, we use the polynomial-based approximation of GELU known as I-GELU where $erf(x)$ is approximated as:

$$L(x) = sgn(x) \cdot [a \cdot (min(|x|, -b) + b)^2 + 1] \tag{3}$$

where $a = -0.2888$, $b = -1.769$, $sgn$ denotes the sign function, and $min$ denotes the minimum function. I-GELU is defined as:

$$I - GELU(x) = \frac{1}{2}x[1 + L(\frac{x}{\sqrt{2}})] \tag{4}$$

However, TFLite does not support the sign function, and the Edge TPU compiler does not support the absolute value function. Therefore, we further revised the GELU approximation and replaced the sign and absolute value functions with $sgn(x) \approx tanh(x \cdot 10^3)$ and $abs(x) \approx x \cdot sgn(x)$, respectively. Thus, we approximate $L(x)$ in (3) as:

$$L(x) = tanh(10^3 x)[a[min(x \cdot tanh(10^3 x), -b) + b]^2 + 1] \tag{5}$$

The proposed I-GELU approximation is supported by both TFLite and Edge TPU. Therefore, in the Transformers, we replace all instances of GELU with our approximation of GELU.

*2) Refactoring Matrix-Matrix Dot Products for FC Layer:* Many of the operations in Transformers are matrix-matrix dot products. Although the matrix-matrix dot product in the self-attention layer is supported by the Edge TPU, it cannot handle the matrix-matrix dot products in the FC layers, as described in the device documentation [16]. To perform matrix-matrix dot products in FC layers, we implement the dot product

operation using convolutions. This can be done as follows: let $A$ be an $m \times n$ input matrix, $B$ be an $n \times k$ weight matrix, and $C$ be the $m \times k$ output matrix such that $A \cdot B = C$. This is a standard matrix-matrix dot product, as shown in Fig. 3 (a). Now consider a convolution layer where we have $k$ convolution kernels, each with the size of $1 \times n$ called $K_{conv}$ (shown in Fig. 3b). We can map the weights from matrix $B$ one-to-one such that $K_{conv}[x] = B^T[x]$. By convolving the kernels $K_{conv}$ across the input matrix $A$ with strides of 1 and no padding, the resulting matrix will be an $m \times k$ matrix identical to the original output matrix $C$ as illustrated in Fig. 3.

Using the aforementioned strategies, we can successfully compile small- and medium-sized Transformers, such as BERT-Tiny to BERT-Medium, on Edge TPU. However, the compilation still fails for larger Transformers such as BERT-Base. Unfortunately, the compiler does not provide detailed information about why the larger models cannot compile, so it is unclear whether the compilation fails due to some fundamental hardware limit in the Edge TPU or if there is an issue with the compiler itself. Regardless, in the next section, we discuss methods to identify the source of the issue and resolve it.

## IV. THE PROPOSED METHODOLOGY TO DEPLOY LARGE TRANSFORMERS ON EDGE TPU

By comparing the architecture of BERT-Medium and BERT-Base (see Table I), we narrow down the possible cause of the compilation failure to the increased hidden size, attention heads, hidden layers, intermediate size, or some combination of these model parameters. Starting with the BERT-Medium architecture, we change one of the model parameters to match BERT-Base until we reproduce the issue. Using this strategy, we identify the two layers that cause the compilation to fail: the inner FC layer and the embedding layer.

Further, we observe that for the inner FC layer, which uses the intermediate size from Tab. I, the model compiles for BERT-Medium using a size of 2048 but does not compile for BERT-Base using a size of 3072. Motivated by this observation, we use a binary search algorithm to determine the maximum size for the inner FC layer that can be compiled on Edge TPU. We find that when followed by the I-GELU activation function, the maximum inner FC layer size for Edge TPU is equal to 2728. Moreover, we find that the maximum size supported varies depending on the type of activation function. For instance, with no activation function

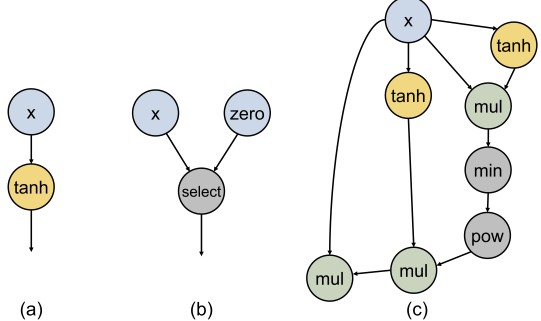

(a)          (b)          (c)

Fig. 4: Computational graphs for (a) $tanh(x)$ (b) ReLU$(x)$ (c) I-GELU$(x)$ activation functions.

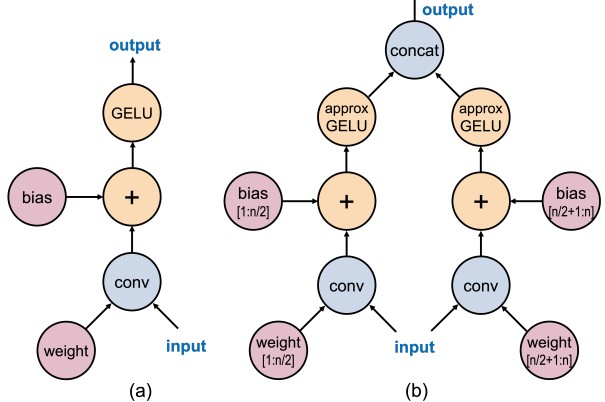

(a)          (b)

Fig. 5: (a) standard convolution-based fully connected layer (b) fully connected layer partitioned across the output dimension.

or using ReLU, Sigmoid, and TanH, the intermediate size can be a maximum of 5376 neurons. This could be due to the computation size of the I-GELU activation function, as it can be seen in its computational graph in Fig. 4c, which can lead to increased memory demands beyond what is available in Edge TPU's PE memory.

To address the aforementioned challenge, we propose partitioning the inner FC layer into two or more equal parts to reduce the size of the operations in the layer. For the BERT-Base model, we partition the inner FC layer into two parts, as shown in Figure 5. By partitioning the layer along the output, we reduce the size of the operation by splitting it into two $m \times n/2$ dot products instead of one $m \times n$ dot product and two $n/2$ I-GELU activations. For the BERT-Base model, this leads to 1536 I-GELU neurons, which is less than the maximum 2728 neurons supported by Edge TPU. At the end, we concatenate these two layers to realize the output. This approach allows the model with the intermediate size of 3072, which is used in BERT-Base, to compile successfully.

Although partitioning the inner FC layer resolves the compilation issue for models with larger intermediate sizes, we still observe that increased size of the embedding layer causes the compilation failure for the model. Therefore, we leverage a similar partitioning mechanism for the embedding layer across the output dimension. Since the embedding layer itself cannot be mapped to the Edge TPU, the model's input for the Edge

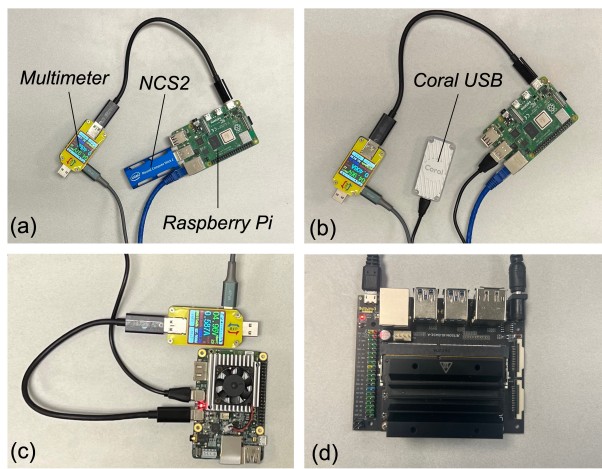

Fig. 6: Experimental setup. (a) Pi + NCS2 (b) Pi + Coral TPU (c) Coral Dev board (d) Jetson Nano.

TPU is the output of the embedding layer. As discussed earlier, here we set the input sequence length to 128; therefore, using BERT-Base for example, the Edge TPU input includes three $128 \times 768$ matrices. Similar to how we partition the FC layer, we partition the embedding layers across the output dimension. This changes the Edge TPU input from three $128 \times 768$ matrices to six $128 \times 384$ matrices.

Using the aforementioned partitioning mechanisms in the inner FC layer and embedding layer, we successfully compile and deploy the BERT-Base and Bert-Large models on Edge TPU. To assess the validity of our approach for other types of transformers, we create a left-right transformer based on the model introduced in [1]. Without any modifications to the model, it fails to compile. However, leveraging our deployment methodology, we can compile and deploy this transformer model on Edge TPU as well, which exhibits the effectiveness of our approach for various architectures and sizes.

## V. EXPERIMENTAL RESULTS

### A. Experimental Setup

After verifying the successful compilation of various-sized Transformer models on Edge TPU, we evaluate its performance against well-known edge AI accelerators existing in the market. In particular, we investigate two experimental setups: *(1) USB accelerators*, where we compare Intel NCS2 (Fig. 6a) with Coral TPU USB accelerator (Fig. 6b), and *(2) Development Boards*, in which we evaluate Coral Edge TPU Dev Board (Fig. 6c) against Nvidia Jetson Nano (Fig. 6d). The USB accelerators are integrated as a co-processor with Raspberry Pi 4. There are different settings required to run the models on each of the edge devices. For the Raspberry Pi 4 and both Coral products, we use TFLite models with fp32 and int8 precision, respectively. For the NCS2, we use OpenVino models with fp16 precision. For the Jetson, we use TensorRT models with fp16 precision. Jetson provides two different operating modes, i.e., low-power mode and Max-N or high-power mode. Here, we use six different BERT models for our experiments: Tiny, Mini, Small, Medium, Base, and

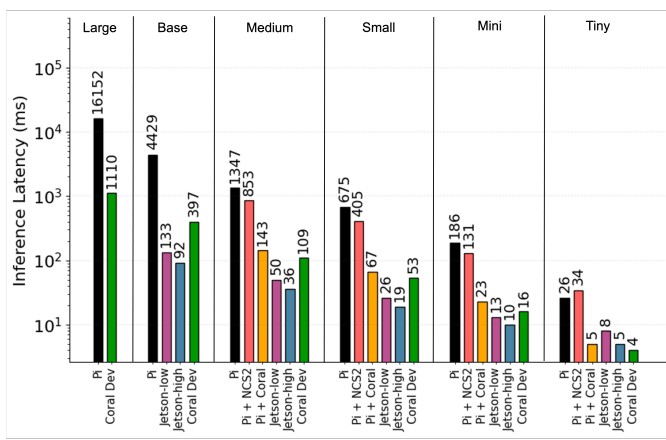

Fig. 7: Inference latency measurements for all models and devices.

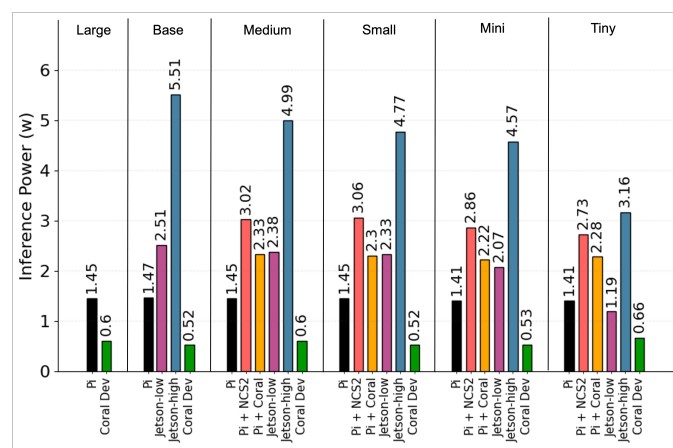

Fig. 8: Dynamic power for all models and devices.

Large. Due to the large size of the BERT-Base and BERT-Large models, we only use development boards to run these models. Also, since Jetson Nano could not compile BERT-Large, we only compared the Coral Dev board with Raspberry Pi for the BERT-Large model.

### B. Inference Latency Measurement

For inference latency measurements, we split the process into three parts: (1) load the model, (2) allocate the tensors depending on the platform, and (3) perform 100 inferences using a subset of the Microsoft Research Paraphrase Corpus (MRPC) dataset [19]. We measure the total time taken for 100 inferences and report the mean inference time for one input sample. Figure 7 shows the inference results for all platforms using the six BERT models. We see that all edge accelerators provide significant speedup over the baseline Raspberry Pi 4 CPU. For the smallest model, BERT-Tiny, Coral Dev board has the fastest inference speed at 4 ms per inference. For the BERT Mini, Small, Medium, and Base models, we see that order of inference speed from least to greatest is as follows: Jetson high power mode, Jetson low power mode, Coral Dev board, Coral USB, NCS2, Raspberry Pi 4. We see that the larger the model is, the bigger the difference is between the coral products and the Jetson. Although we do not report the model load and allocation times, it is important to note that the Coral Dev board, Coral USB, Jetson, and Raspberry Pi all take less than 10 seconds to load and allocate BERT-Medium, while the NCS2 takes over 10 minutes to load and allocate the same model.

*1) USB Accelerators:* For the USB accelerators, both NCS2 and Coral USB accelerator show improvement over the baseline Raspberry Pi 4, except for the case of NCS2 and BERT-Tiny. For BERT-Tiny there is an improvement of $0.76\times$ for NCS2 and $5.2\times$ for Coral USB accelerator. For BERT-Medium, we observe approximately $6\times$ reduction in inference latency for Coral USB accelerator compared to NCS2.

*2) Development Boards:* Both development boards offer significant speedups compared to the Raspberry Pi 4. For the BERT-Tiny model, we observe $3.2\times$ and $5.2\times$ improvement

over the baseline model using Jetson low and high power modes. For Coral Dev board, we have $6.5\times$ improvement over the baseline model. Performing the same comparison for the BERT-Base model, we have $33\times$ and $48\times$ improvement over the baseline model for Jetson with low and high power, respectively. For Coral Dev board, we observe $11\times$ improvement of the baseline model. For smaller models, Coral Dev board is slightly faster than the Jetson, but for larger models, Jetson is up to $4.3\times$ faster. The faster inference of Jetson, however, is achieved at the cost of significantly more chip resources and increased power consumption, as discussed in the next subsection.

### C. Inference Power Measurements

We use MakerHawk UM34C USB multimeter to measure the power dissipation of all devices, except for the Jetson, which has three internal sensors for measuring the input, CPU, and GPU powers. To obtain the average power consumption, we run each model on each platform for five minutes and record the corresponding power profiles. Figure 8 shows the dynamic power measurements for all the models and platforms. Coral Dev board has the lowest power consumption across all experiments. As shown in the figure, the power consumption remains roughly unchanged across various models for all the platforms, except for Jetson's power consumption, which grows with the model size.

*1) USB Accelerators:* For the BERT-Tiny and BERT-Medium models, NCS2 and Coral USB accelerator consume nearly $1.9\times$, and $1.6\times$ more power than Raspberry Pi 4 alone. Coral USB consumes 1.3x less power than NCS2.

*2) Development Boards:* For the BERT-Tiny model, Coral Dev board achieves $2.1\times$ reduction in power compared to Raspberry Pi and a $4.8\times$ improvement over Jetson in high-power mode. For the BERT-Base model, Coral Dev board realizes a $9.4\times$ and $4.8\times$ power reduction compared to Jetson in high-power and low-power modes, respectively. Finally, for the BERT-Large model, Coral Dev board can achieve $2.4\times$ reduction in power dissipation compared to Raspberry Pi.

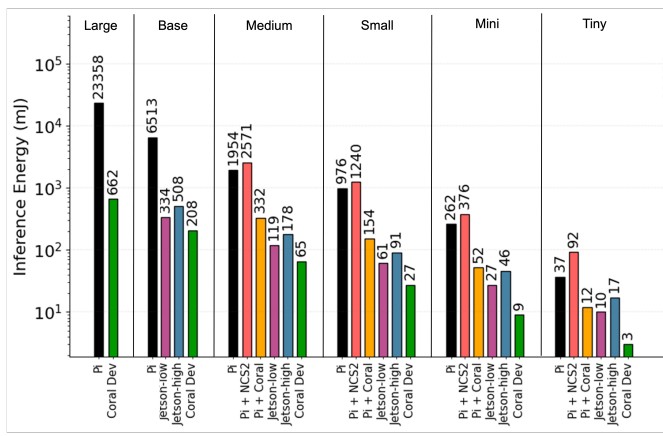

Fig. 9: Inference energy for all models and devices.

### D. Inference Energy

In Fig. 9, we compare the results for inference energy. Aside from NCS2 we see that all accelerators significantly improve inference energy over baseline Raspberry Pi.

*1) USB Accelerators:* For the USB accelerators, we compare the BERT-Medium model. For The NCS2, the inference energy is 1.3× worse than the Raspberry Pi 4 with no acceleration. Interestingly, the Coral USB accelerator provides a 5.9× and 7.75× improvement in inference energy compared to the Raspberry Pi alone and Raspberry Pi with NCS2, respectively.

*2) Development Boards:* Compared to Raspberry Pi, Coral Dev board provides a 12× decrease in inference energy for the BERT-Tiny model. When compared to Jetson-low and Jetson high, Coral Dev board provides 3× and 6× improvements, respectively, for the same model. For the BERT-Base model, Coral Dev board is 1.6× and 2.5× more efficient than Jetson-low and Jetson-high. Furthermore, Coral Dev board is 31× more efficient than Raspberry Pi for the same model. Finally, for the BERT-Large model, Coral Dev board achieves a notable 35× energy saving compared to Raspberry Pi.

## VI. CONCLUSION

This paper provides a methodology to deploy Transformer models on Edge TPU accelerators by identifying the layers in Transformers that are not supported by Edge TPU and refactoring their computational graph. We provide an extensive comparison of the leading edge devices on the market for Transformer models. Our methodology can deploy various Transformer architectures on the Coral Edge TPU and achieves real-time inference while maintaining the lowest energy consumption of the edge devices. We show that by adopting our approach for the Coral USB Accelerator, inference for medium-sized Transformers can be accelerated up to nearly 10× while consuming 6× less energy. Further, for large Transformers, our approach may be the only viable approach due to the memory constraints associated with other edge devices.

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
