# OpenReview forum: "Efficient Deployment of Transformer Models on Edge TPU Accelerators: A Real System Evaluation"
_iscaconf.org/ISCA/2023/Workshop/ASSYST — ASSYST Oral_

### Official Review · Reviewer_Y2KS · 2023-05-05
**Review for Efficient Deployment of Transformer Models on Edge TPU Accelerators: A Real System Evaluation**

**Rating:** 7
**Confidence:** 3

**Review:**

This paper presents a case study evaluation of running the BERT model on the Coral Edge TPU accelerator. The work describes the methodology and steps required to successfully deploy a large transformer on the low-cost portable accelerator, and it also compares their optimization for Coral TPUs against other AI accelerators. The paper characterizes some of the challenges across the design space of transformer sizes, including customized implementation of Keras layers to achieve performance, and it evaluates latency, energy, and power.

**Review (Strengths/Weaknesses):**

### Strengths
- I appreciated the detailed presentation of what it takes to do a deployment of a large-scale model on a constrained accelerator.
- The work documents a number of challenges and their solution, producing a useful data point for others interested in similar deployments.
- Although the article is presented primarily to focus on deploying BERT on Coral TPUs, it is notable that the evaluation compares against functional deployments on multiple other low-cost accelerators.

### Weaknesses
- It is a bit unfair to evaluate the other accelerators at fp16 against int8 on the Coral Dev Board, and the actual bitwidths evaluated could be presented in more clearly. I interpreted the text as saying the Pi + Coral was running at fp32 and the Coral Dev ran at int8, but it was unclear and the discussion in the results didn't acknowledge the impact of bitwidth on latency/memory consumption/power.
- A paper with this type of scope could have included more candid discussion of the actual deployment experience, even if the topics are implied during the methodology sections -- for instance, included as a Limitations and Future Work section. Were the challenges more in the compiler frameworks or the device resource constraints? Were there any tradeoffs between USB accelerators and dev boards that proved most limiting?

### Additional questions, comments
- I understand the focus was on the Coral TPU's idiosyncracies when building BERT, but I also found myself interested in the constraints on the other accelerators as well. For instance, why was the Jetson Nano unable to compile BERTLarge, was it memory constraints?
- Nits: Coral is often not capitalized correctly (Page 1, Page 6)

**Reviewer Expertise:**

Knowledgeable: I used to work in this area and/or I try to keep up with the literature but might not know the latest developments.

---

### Official Review · Reviewer_GMc6 · 2023-05-06
**Efficient Deployment of Transformer Models on Edge TPU Accelerators: A Real System Evaluation**

**Rating:** 7
**Confidence:** 4

**Review:**

Summary:
* The authors propose recomposition of the computation graphs for large, medium and small transformer based networks to map it onto edge devices, specifically on the edge TPU.
* Among the networks the authors suggest refactoring GELU operations, and matrix-matrix dot product using existing operations supported by TF-Lite.
* For large transformer layers the proposed approach involves splitting the large FC operations into sequential parts to ensure the code compiles
* Resulting changes leads to significant improvements in latency and energy on edge TPU when compared with an equivalent implementation on a general edge platform like Raspberry-Pi

**Review (Strengths/Weaknesses):**

Strengths
* Clear writing which is easy to follow
* Implementation of real hardware
* The optimizations specific various sizes of networks are relevant

Additional points to improve
* Comparison with existing baselines using TVM or other NN compiler backend
* Providing performance results on relevant workloads

**Reviewer Expertise:**

Knowledgeable: I used to work in this area and/or I try to keep up with the literature but might not know the latest developments.

---

### Official Review · Reviewer_wkVV · 2023-05-10

**Rating:** 6
**Confidence:** 4

**Review:**

1. Research problem(s) and research contribution(s) are unclear in the Abstract and Introduction sections. E.g.,
* The following sentence should be well-explained: “Here, we modify the computational graph to allow the model to run on the Edge TPU while remaining functionally identical to the original model.”
2.	The manuscript contains a few typos. E.g.,
* Introduction: “(_[10] provides an overview …)” -> “([10] provides an overview …)”


**Review (Strengths/Weaknesses):**

## Strengths
*	It is a relevant topic for the conference and the paper's message is well-supported.
*	Providing a nice summary of existing requirements for running a DL model on Coral Edge TPU.
*	Increasing the applicability of Edge TPUs in supporting diverse DL models by proposing some techniques to tackle the limitations. I can imagine several industrial applications can benefit from this paper.
## Weaknesses
*	There is room for improvement in the writing of the paper.
*	It is hard to say the proposed techniques are novel. Plus, there is no comparison with related studies (indeed there is no related work section).
*	As always, to guarantee the reproducibility of results, I highly recommend authors release the code of this paper.


**Reviewer Expertise:**

Expert: I have written one or more papers on this topic and/or I currently work in this area.